# Inadequate Intake of Choline and Essential Fatty Acids in Latin American Childbearing-Age Women as a Regional Pre-Conceptional Disadvantage: ELANS Results

**DOI:** 10.3390/nu16183150

**Published:** 2024-09-18

**Authors:** Marianella Herrera-Cuenca, Martha Cecilia Yépez García, Lilia Yadira Cortés Sanabria, Pablo Hernández, Guillermo Ramírez, Maura Vásquez, Yaritza Sifontes, Georgina Gómez, María Reyna Liria-Domínguez, Attilio Rigotti, Mauro Fisberg, Irina Kovalkys, Maritza Landaeta-Jiménez

**Affiliations:** 1Centro de Estudios del Desarrollo, Universidad Central de Venezuela (CENDES-UCV), Caracas 1053, Venezuela; 2Fundación Bengoa, Caracas 1053, Venezuela; yaritza2sifontes@gmail.com (Y.S.); mlandaetajimenez@gmail.com (M.L.-J.); 3Department of Nutrition and Health, Framingham State University, Framingham, MA 01702, USA; 4Colegio de Ciencias de la Salud, Universidad San Francisco de Quito, Quito 170901, Ecuador; myepez@usfq.edu.ec; 5Departamento de Nutrición y Bioquímica, Pontificia Universidad Javeriana, Bogotá 110231, Colombia; ycortes@javeriana.edu.co; 6Escuela de Nutrición y Dietética, Facultad de Medicina, Universidad Central de Venezuela, Caracas 1053, Venezuela; pablo.i.hernandez@ucv.ve; 7Área de Postgrado en Estadística, Facultad de Ciencias Económicas y Sociales, Universidad Central de Venezuela, Caracas 1053, Venezuela; guillermo.ramirez.ucv@gmail.com (G.R.); mauralvasquez@gmail.com (M.V.); 8Departamento de Bioquímica, Escuela de Medicina, Universidad de Costa Rica, San José 11501-2060, Costa Rica; georgina.gomez@ucr.ac.cr; 9Department of Nutrition and Health, Instituto de Investigación Nutricional, La Molina, Lima 15026, Peru; rliria@iin.sld.pe; 10Centro de Nutrición Molecular y Enfermedades Crónicas, Departamento de Nutrición, Diabetes y Metabolismo, Escuela de Medicina, Pontificia Universidad Católica, Santiago 8330024, Chile; arigotti@med.puc.cl; 11Centro de Excelencia em Nutrição e Dificuldades Alimentaes (CENDA), Instituto Pensi, Fundação José Luiz Egydio Setubal, Hospital Infantil Sabará, São Paulo 01228-200, Brazil; mauro.fisberg@gmail.com; 12Departamento de Pediatria, Universidade Federal de São Paulo, São Paulo 04023-061, Brazil; 13Carrera de Nutrición, Facultad de Ciencias Médicas, Pontificia Universidad Católica Argentina, Buenos Aires C1107 AAZ, Argentina; ikovalskys@gmail.com

**Keywords:** choline, essential fatty acids, women of childbearing age, diet, Latin America, ELANS, dietary intake

## Abstract

Background/Objectives: Choline and essential fatty acids (EFA) are vital for fetal brain development, supporting pregnancy, and maintaining hormonal balance. They also promote overall health. The childbearing years present a window of opportunity to increase the intake of these key nutrients and develop healthy dietary habits. The aims of this study were to evaluate the intake of choline and EFA in women of childbearing age (15–49 years old), identify their food sources and determine if supplements containing choline and EFA were available across the Estudio Latinoamericano de Nutrición y Salud (ELANS) countries. Methods: Survey data were collected for the ELANS, including participants from Argentina, Brazil, Chile, Colombia, Costa Rica, Ecuador, Peru, and Venezuela (*n* = 9218; 15–65 years old). Women of childbearing age were extracted from the largest database (*n* = 3704). Results: In general, choline intake was inadequate in all countries, while EFA intake was normal or above requirements. Chile had the lowest intake of choline, and Colombia had the highest. The results showed that some countries had more inadequate choline intake than others. Consuming a larger quantity of eggs helped reduce choline inadequacy, as did including eggs and fish in the diet. The intake of EFA, including ALA, EPA, and DHA, showed variability. The contributions of EPA and DHA were lower than that of ALA, and the results differed by age group. Conclusions: choline intake is inadequate, and EFA intake is variable among women of childbearing age in the ELANS study. More awareness and education are needed to achieve better intake of these nutrients.

## 1. Introduction

Choline is a precursor of several key biomolecules, such as acetylcholine, phosphatidylcholine, sphingomyelin, and betaine. Each of those is involved, respectively, in (1) neurotransmission that regulates early brain development and the function of attention, (2) as a component of biological membranes, (3) as a myelin constituent, and (4) as a methyl donor that can engage in epigenetic marks through the DNA methylation processes, synthesis and repair [1,2,3]. Recently, it has gained attention due to its role as a precursor molecule for the synthesis of phospholipids and the methyl-donor betaine [1], and it also has been used to improve physical performance in adults; however, its effects on the muscles are unclear [4].

The roles of choline during fetal development are numerous, and its requirements during pregnancy and lactation are high [2] and its relevance to health includes supporting normal brain development, acting as a protective factor when alcohol damage and infections occur and ameliorating intellectual disabilities by improving neural—cognitive functioning and memory [3,5].

Due to the importance of EFA in the neurodevelopmental processes and as a component of important key derivatives that are needed by the body, such as vitamin D, the evaluation of the intake of EFA in pregnant and lactating women is as relevant as choline intake.

Despite numerous research on this topic in animal models, including intake and its effects when deficient, very few studies in humans have been conducted on choline and EFA and the impact of adequate/inadequate intake in brain, cognitive and neurodevelopment in the human offspring [6]. A study on childbearing age and pregnant women’s consumption of EFA (DHA and EPA) in the United States reported an inadequate intake of both nutrients, particularly in women belonging to disadvantaged communities, highlighting the concerns of deficiencies in socioeconomically vulnerable women [6].

In addition, from those few studies, the evidence shows that an amount of choline that is more than double the requirements is needed during the third trimester of pregnancy to improve the attention and focus of school-aged children [7]. Data obtained by the Viva Cohort study in Massachusetts showed that maternal choline intake within the recommended range during pregnancy was also associated with better memory function in children at seven years old compared to those whose mothers had an intake of 50% of the requirements [8].

The synthesis of choline, as well as vitamin D and docosahexaenoic (DHA), can occur in the body; however, the produced amounts are insufficient to support the needs. Therefore, it is important to have an adequate intake to support this process [2,6].

Much of the relevance of choline and other nutrient intake is usually highlighted for pregnant and lactating women; much less is said about the adequacy of the intake in women of childbearing age. However, while there is significant knowledge and education on the benefits of folic acid or iron intake during the pre-conceptional period, very little is known about choline and EFA intake during this period, and consequently, information that is given to women is scarce promoting a knowledge gap regarding these nutrients [9,10].

Assuming that eating habits are routine and women’s adequate pre-conceptional eating is a part of the important behavior that directs a pregnancy toward a successful ending, we need to understand what the consumption trends are in this period and, therefore, determine whether women are well prepared to embrace pregnancy or not [11,12]. In addition, parental and particularly maternal pre-conceptional overall lifestyle can influence the long-term risk of future generations in terms of later development of chronic diseases and other morbidities, as stated in the Developmental Origins of Health and Disease concept [13].

In Latin America, access to adequate nutrition can be challenging for an important proportion of the population, including women of fertile age. The Food and Agriculture Organization of the United Nations (FAO), in its panorama of food security for Latin America and the Caribbean [14], states that women are more food insecure than men across the region, therefore exposing them to deficiencies that potentially will continue if getting pregnant.

In addition, it is important to draw the attention of the general population, and of women of childbearing age, pregnant and lactating women, to identify the sources of choline and EFA in the diet and what average intake may be in different populations around the world. Further research on choline and EFA in Latin America is a key aspect of addressing the next generation’s nutritional well-being, which makes this research a key aspect of the evaluation of the Latin American Study of Nutrition and Health (ELANS) by its acronym in Spanish (Estudio Latinoamericano de Nutrición y Salud) data.

More epidemiological studies are needed to better understand the intake of different nutrients. ELANS is an epidemiological cross-sectional study that gathered information on women of childbearing age intake of choline and EFA, which is a window of opportunity during the lifespan that can provide a different perspective to approach interventions. This study represents a research effort to fill the existing knowledge gap in the epidemiological field.

To the best of our knowledge, this is the first study that addresses choline and EFA intake in eight countries within the region, particularly on women of fertile age; thus, our aim was to study the choline and EFA intake in childbearing-age women, identify the food sources, learn if supplements that contained choline and EFA existed across the studied countries, and identify what potential policies would be adequate to address the deficiencies.

## 2. Materials and Methods

### 2.1. Study Sample

The data for this study were sourced from ELANS, a multicenter cross-sectional research project carried out across eight Latin American countries, including Argentina, Brazil, Chile, Colombia, Costa Rica, Ecuador, Peru, and Venezuela. Conducted over a one-year period from September 2014 to August 2015, this study focused on assessing the dietary intake and physical activity levels of individuals living in households. The study used a representative sample of urban populations, where 80–90% of the population resides in these countries.

The sample was obtained using a multistage probability sample stratified by geographical location, sex, age, and socioeconomic status. A total of 10,134 participants were eligible for the ELANS study; however, due to refusal, 9680 participants comprised the sample. Only those with complete data who answered the socioeconomic questionnaire and two 24 h dietary recalls were included in the ELANS study (*n* = 9128 participants). In this research, we analyzed data from women of childbearing age, defined as those aged between 15 and 49 years; therefore, men and women over 49 years of age were excluded from the analyses. The final sample was comprised of 3704 women aged 15 to 49 years old (Figure 1).

### 2.2. Sociodemographic Variables

Women were grouped into three age categories (15–19 years, 20–34 years, and 35–49 years) according to WHO definitions [15] of women of childbearing age. Socioeconomic status (SES) was assessed using a country-dependent questionnaire format in accordance with legislative requirements or established local standards. The SES classification included three categories (low, medium, and high) based on national indexes specific to each country. This methodology was utilized to determine the mean per-person income of households and compare it with established thresholds for Latin Americans, as outlined by Fisberg et al. [16]. The categories of none or basic education, high school, and a Bachelor’s degree were utilized to distinguish educational levels. Marital status was grouped into single, married or coupled, and divorced or widowed.

### 2.3. Dietary Data

Dietary intake information was assessed using two non-consecutive 24 h recalls with two face-to-face household visits, with an interval of up to eight days between visits, with a focus on capturing day-to-day variation in food consumption by including both weekdays and weekend days in the recalls. The Multiple-Pass Method was employed to ensure a detailed recording of all foods and beverages consumed on the previous day. Portion sizes were estimated using photographic albums of common foods and household utensils. Local and traditional foods were harmonized using the USDA composition table for nutritional equivalence. The food intake data obtained from the recalls were standardized and converted into grams and milliliters by qualified nutritionists in each country. The collected data were transformed into nutrients using Nutrition Data System for Research software (NDS-R, Minnesota University, Minneapolis, MN, USA version 2013) [17].

All food items were classified into one of the eighteen specific food groups based on their nutritional content. The food groups were as follows: (1) cereals, (2) tubers, (3) legumes, (4) dairy, (5) eggs, (6) beef, (7) poultry, (8) fish, (9) pork, (10) other meats, (11) vegetables, (12) fruits, (13) nuts and seeds, (14) butter, (15) margarine, (16) oils, (17) other fats, and (18) miscellaneous. Only the top five food group contributors to each nutrient were displayed in the results.

#### 2.3.1. Choline and Essential Fatty Acids Intake

The Multiple Source Method (MSM), a web-based statistical modeling approach recommended by the European Prospective Investigation into Cancer and Nutrition (EPIC), was implemented to transform 24 h recall-derived individual intake into usual daily intake distributions [18].

The study specifically focused on two groups of EFA: omega-3 (w-3) and omega-6 (w-6). Among the w-3 fatty acids, Alpha-Linolenic Acid (ALA), Eicosapentaenoic Acid (EPA), and Docosahexaenoic Acid (DHA) were examined. In contrast, w-6 contains Linoleic Acid (LA) and Arachidonic Acid (ARA).

Through the employment of the MSM, the study was capable of estimating the usual intake of choline and essential fatty acids at the population level while taking into account within-person variance. Moreover, the typical intake of choline and essential fatty acids was individually assessed for each country, taking into consideration the variations in dietary habits among Latin American populations.

#### 2.3.2. Choline and Essential Fatty Acids Adequacy

The intake levels of choline and EFA were assessed and evaluated against the US Institute of Medicine (IOM) Dietary Reference Intakes (DRI) standards [19], which stipulate a daily intake of 425 mg for choline, 1.1 g for ALA and 11 or 12 g for LA in women between the ages of 15 and 18, and 19 to 49, respectively. The adequacy of each individual’s nutrient intake was calculated using the following formula: the observed intake value was divided by the DRI and multiplied by 100. The subject was classified as normal if the resulting number fell between 90% and 110%. If the number was less than 90%, the individual’s nutrient intake was deemed inappropriate because of deficiency. Conversely, if the number was greater than 110%, the individual’s intake was deemed inappropriate because of excess.

It should be highlighted that ALA and LA are considered EFA because humans cannot synthesize them. On the other hand, ALA is metabolized to EPA and subsequently to DHA, while LA is the precursor of ARA. Thus, EPA, DHA, and ARA are not considered essential fatty acids as they can be synthesized endogenously [20]. For this reason, the National Academy of Medicine, the former Institute of Medicine, has not established Dietary References Intakes (DRIs) for EPA, DHA, and ARA. The IOM only recommended that EPA and DHA contribute 10 percent of the total omega-3 fatty acid intake, which should be approximately 160 mg/day [19]. Therefore, a percentage of adequacy cannot be provided for EPA, DHA, and ARA.

### 2.4. Choline and Essential Fatty Acids within Prenatal Multivitamins-Supplements

To identify if choline and EFA existed as part of the prenatal multivitamin–mineral–nutrient supplements available in the eight ELANS countries, the research team agreed to find the three most used prenatal supplements through experts and health professionals. After gathering the names, carefully reading the labels and identifying the components named in each supplement, a categorization according to: “choline included” and “EFA included” was performed to determine whether choline and essential fatty acids were available in their composition.

### 2.5. Statistical Analysis

Descriptive statistics analysis was computed for continuous variables as means, standard error of the mean (SEM), and minimum and maximum values. Categorical measures are presented as counts and percentages. The intake values in adequacy were compared using the Mann–Whitney or Kruskal–Wallis test after the normality had been tested by the Skewness and Kurtosis test. In addition, chi-square analysis was used to evaluate possible associations. These analyses were performed for choline as well as for EFA.

To evaluate the impact of different factors that could affect choline adequacy, a classification tree (CT) was applied to determine differences between countries and the food sources that result in inadequate choline intake. This technique selects predictors that produce differentiated groups in accordance with adequacy levels, which is important for the purpose of this study due to the relevance of choline and because, in general, there is less information on choline intake. In all cases, tests were performed with a statistical significance level of 0.05. Microsoft Excel^®^ 2016 software and the IBM SPSS^®^ version 25 statistical packages were used for data loading and analysis.

### 2.6. Ethics

The ELANS protocol, which was registered at Clinical Trials (#NCT02226627) and approved by the Western Institutional Review Board (#20140605), also received approval from the ethical review boards of the participating institutions. Participants provided informed consent for inclusion in the country-level study, and their confidentiality was maintained using identification codes instead of names.

## 3. Results

### 3.1. Sociodemographic Characteristics

An analysis of data was performed on women of childbearing age, which was defined as individuals aged between 15 and 49 years. The sociodemographic characteristics of the participants, including age (SES), educational level, and marital status among countries, were studied by applying a chi-square test to evaluate the heterogeneity of the variables whose profiles are presented in Table 1. The distribution of women according to age varied across the countries included in this study (*p* = 0.043). Costa Rica and Ecuador had a higher proportion of younger women (17.8% and 17.3%, respectively), while Argentina and Brazil had a relatively larger population of older women of childbearing age (41.5% and 40.6%, respectively). Venezuela had the highest proportion of women in the low socioeconomic level (76%), whereas Peru, Costa Rica, and Ecuador showed a prevalence >10% in the high socioeconomic level (*p* < 0.001). The educational level of Venezuela was considerably higher than that of the other countries (*p* < 0.001). A Bachelor’s degree was reported by 20.7% of Venezuelan women, nearly twice the mean proportion in the total ELANS group. On the other hand, Ecuador had the highest proportion of women with no or basic educational level (83.6%). Most women across all countries were married or in a relationship, with Ecuador having the highest percentage (57.7%), followed by Peru (57.5%). Argentina had the highest percentage of divorced or widowed women (12.3%), while Venezuela and Chile had a higher proportion of single women (48.2% and 47.2%, respectively) (*p* < 0.001).

### 3.2. Choline Intake

Table 2 displays the choline intake and adequacy of the sample with deficient, normal, or excessive consumption. The average daily choline intake of the study participants was 305 mg/day, which was 71.8% of the IOM-recommended intake of 425 mg/day (adequacy); consequently, women in this population are in deficit of this nutrient by 28 mg/day. As a result, 79.8% of the participants were deficient in choline consumption. All countries follow this trend, especially Chile, which had the lowest intake (238.6 mg/day) and the highest proportion of deficient consumption (95.4%). Additionally, Colombia had the highest intake (373.1 mg/day) and the highest proportion of excessive intake (19%). Despite this, more than half of the Colombian women (55.2%) had deficient intake. No significant differences in choline consumption parameters were observed between age groups (*p* = 0.099), socioeconomic status (*p* = 0.608), educational level (*p* = 0.072), and marital status (*p* = 0.057).

Figure 2 provides a comparison of the choline intake (mg/day) of the five most contributing food groups in the ELANS. According to the data, eggs were found to be a significant source of choline in most countries, except Brazil, where beef was the main source (*p* < 0.001). Colombia reported the highest per capita intake of choline from eggs, at 154,02 mg/day. In contrast, poultry consumption exhibited more consistent levels of choline intake across the countries studied. Notably, Peru had the greatest contribution to choline intake (77.92 mg/day) in the poultry group.

Fish and pork consumption exhibited variability, thus having an impact on choline intake levels among Latin American countries. Certain countries, such as Peru (39.4 mg/day) and Brazil (31.82 mg/day), relied heavily on fish as a source of choline. 

In contrast, countries like Argentina had a lower consumption level (6.63 mg/day) in the fish group. Pork consumption also showed variations in choline intake, with some countries such as Costa Rica (37.2 mg/day) and Ecuador (32.2 mg/day) reporting higher contribution levels. However, countries like Peru had the lowest contribution (14.12 mg/day).

### 3.3. Classification Tree

To identify the principal sources of food that explain differences among countries related to the adequacy of Choline’s intake, a classification tree (CT) is presented (Figure 3). In this context, the inadequate by deficit group was represented by “<90%”, while the normal and excessive groups were merged and represented by “>90%”.

The CT implemented revealed a primary division into two groups of countries: Group 1, comprising Argentina, Colombia, and Ecuador (66.7% deficient) and Group 2, comprising Brazil, Chile, Costa Rica, Peru, and Venezuela (86.9% deficient). The second division in both groups was established by the consumption of choline coming from the eggs (mg/day), suggesting that an increase in egg intake helps reduce choline insufficiency. The third division considered egg and fish consumption, as well as the country, as relevant variables. A significant increase in the adequacy level of choline was observed in women who included a larger quantity of eggs or fish in their diets. Furthermore, Chile and Venezuela were classified as countries which, despite having a good contribution of choline from eggs, had a higher proportion of people with deficient intake of this nutrient.

### 3.4. Essential Fatty Acids Intake

Regarding omega-3, Table 3 provides information on the consumption parameters of alpha-linolenic acid (ALA) among women of childbearing age. Additionally, Table 4 highlights the intake of eicosapentaenoic acid (EPA), docosahexaenoic acid (DHA), and the total amount of omega-3 fatty acids (as the sum of ALA + EPA + DHA). As expected, the contributions of EPA and DHA were lower than those of ALA; therefore, their values are presented in milligrams (mg). In all cases, the contribution of DHA was greater than that of EPA. The overall average consumption of ALA among the participants was recorded at 1.38 g, with an adequacy percentage of 125.5%; therefore, 55.6% of the sample exhibited excessive ALA intake. Table 4 shows that the mean contribution of EPA + DHA was less than 100 mg, so the total omega-3 consumed was 1.47 g.

Argentina and Chile exhibited a lower average ALA consumption of 0.98 g and 1.01 g, corresponding to an adequacy level of 89.1% and 92.2%, respectively. They also had the highest percentage of participants with deficient intake at 57.6% and 55.7%, respectively (*p* < 0.001). In contrast, Venezuela exhibited the highest ALA average consumption at 1.72 g, with an adequacy level of 156.3% (*p* < 0.001). Due to this, Venezuela had the lowest percentage of individuals with deficient intake at 11.9%, on par with the highest proportion of participants with excessive intake levels at 76% (*p* < 0.001). This trend was similar to that of the total w-3 values (Table 4). Regarding EPA and DHA, Ecuador had the highest intake of both fatty acids (37 mg and 113.5 mg, respectively), while for EPA, the lowest values were for Argentina, Brazil, and Chile (<18.5 mg). For DHA, the lowest value was found in Brazil (59.3 mg).

These findings indicate that the consumption of ALA, and consequently of total w-3, varies among different age groups, with statistically significant differences (*p* < 0.001), but not for the EPA and DHA intake (Table 4). Specifically, the 35–49-year-old age group exhibited lower ALA intake than the younger group, with an average difference of approximately 200 mg between the two groups. Furthermore, the data reveal that more than half of the women in all three age groups had excessive ALA intake, with the highest percentages identified in the younger age categories (*p* < 0.001). The analysis also indicated that there was no statistically significant difference in w-3 and constituent intake across the different socioeconomic status groups (*p* > 0.050).

There was a slight increase in ALA, total w-3 intake, and ALA adequacy as education level increased (*p* < 0.001), but this was not observed for EPA and DHA (*p* > 0.050). Individuals with a Bachelor’s degree had the highest ALA intake and adequacy percentage (1.48 g and 134.9%, respectively). Women with a Bachelor’s degree had the lowest percentage of deficit in adequacy within the diet (21.1%) and the highest proportion of excessive intake (55.6%). In comparison, those with no education or basic education had the highest ALA percentage of deficiency (32.6%) and a lower proportion of excessive diet (50.3%).

The data suggests a contradictory difference in the omega-3 fatty acids by marital status (*p* < 0.001). While single individuals have the highest average intake of ALA and total omega-3 (1.42 g and 1.51 g, respectively), divorced or widowed individuals have the lowest (1.31 g and 1.41 g, respectively). However, it was quite the opposite for EPA and DHA, with higher intake values in divorced or widowed women (23 mg and 75.4 mg, respectively) in comparison with single women (20.6 g and 67.1 g, respectively).

The five food groups that made the greatest contribution to the consumption of total omega-3 fatty acids are illustrated in Figure 4. These groups were oils, fish, cereals, other fats, and margarine, with notable variations between countries (*p* < 0.001). In most countries, vegetable oils were the primary source of w-3 fatty acids, with the exception of Ecuador, where fish was the primary dietary source (0.61 g). In Argentina, other types of fats (such as animal fat, fatty dressings, and condiments) and cereals were the primary sources (0.3 g and 0.28 g, respectively). Margarine made a greater contribution in Venezuela (0.37 g) than in the rest of the countries.

In relation to omega-6, Table 5 provides information on the consumption parameters of linoleic acid (LA) among women of childbearing age. Additionally, Table 6 highlights the intake of arachidonic acid (ARA) and the total amount of omega-6 fatty acids (as the sum of LA+ARA). As expected, the contributions of ARA were lower than those of LA; therefore, the ARA values are presented in mg. The overall average consumption of LA among the participants was 13.91 g, with an adequacy percentage of 125.5%. A little less than half (49.7%) of the sample exhibited excessive LA intake. The mean contribution of ARA was 141.9 mg, so the total omega-6 consumed was 14 g. 

Chile exhibited the lowest levels of LA, LA adequacy, ARA, and total w-6 intake, whereas Ecuador displayed the highest values (*p* < 0.001). Therefore, Chile was the country with the highest percentage of deficient intake (47.2%), while Ecuador had the highest percentage of excessive intake (73.5%). Similarly, for w-3, higher w-6 intake levels were observed in women between 15 and 19 years old than in those between 35 and 49 years old (*p* < 0.001). These data show a trend of increasing percentage of Deficient weight as age increases from the younger (18.4%) to older (36.6%) age groups. The intake parameters of w-6 turned out to be independent of the SES and the education level of the women (*p* > 0.050). The marital status was significantly different only in the LA adequacy percentage and in the ARA intake. Divorced or widowed women had a lower intake than single women (*p* = 0.004).

The five food groups that made the greatest contribution to the consumption of total omega-6 fatty acids are illustrated in Figure 5. These groups were oils, cereals, other fats, poultry and margarine, with some variations between countries (*p* < 0.001). In all the countries, vegetable oils were the primary source of w-6 fatty acids.

In Argentina, cereals and other fats had a higher contribution to the w-6 intake. Poultry was prominent in the w-6 contribution of the Peruvian women, while margarine was the second-highest contributor of w-6 in Brazil.

To assess the imbalance in the intake of EFA, the w6/w3 ratio was calculated—it is reported as a Appendix A. The main results are that Venezuela exhibited the lowest regional value, with a 95% confidence interval ranging from 6.98 to 7.18. Peru, Colombia, and Brazil, countries with relatively higher values than Venezuela, had ratios ranging from 8.04 to 8.20 with 95% confidence. Costa Rica, Chile, Ecuador, and Argentina demonstrated progressively increasing mean ratios (95% CIs: 9.19–9.67, 10.74–11.64, 12.29–13.39, and 17.38–18.50, respectively), with statistically significant differences between all pairs of countries.

### 3.5. Available Choline and Essential Fatty Acids in Supplements

Table 7 shows the three different brands most prescribed and available as prenatal supplements in the eight countries that constitute ELANS. The analysis revealed that none of the products in Argentina, Brazil, Chile, Colombia, Costa Rica, Peru, or Venezuela contain choline or omega-6. However, at least one brand in each country declares the presence of omega-3 in their supplements. Most brands across the countries declare the DHA/EPA content, except for Venezuela, where only the total omega-3 content is specified. Colombia stands out as the country with all three brands declaring the presence of omega-3, followed by Chile, Brazil, and Venezuela, with two brands each declaring omega-3. Additionally, Argentina, Costa Rica, and Peru are noted as countries where the DHA content in prenatal supplements is below 200 mg.

## 4. Discussion

In this study, a prevalence of 80% of women between 15–49 years old, with inadequate intake of choline, was found (below 90% of the recommended intake established). An astonishing majority of women have inadequate consumption of this nutrient across the eight studied countries. As reported in other studies, choline intake is largely deficient in pregnant women, which has deleterious effects on fetal brain development and the ability to concentrate and visual memory for school-age children [3,8]. Also, because of this poor choline intake pattern during pre-conceptional years, one can easily understand that when these women eventually become pregnant, they will likely continue these eating habits unless an intervention is strategically planned [20].

However, the intake of EFA can be disaggregated twofold: one regarding ALA and LA and the other regarding DHA, EPA, and ARA. While the ALA and LA showed adequacies of 125%, the intake of DHA + EPA was below 100 mg per day, below the suggested 160 mg per day. Again, DHA and EPA are two long-chain EFA associated with cardiovascular health, joint and eye health, and fetal brain development. For this study, the researchers agree with using the former IOM recommendations for choline and EFA because there are no recommended values in Latin America. However, other authorities recommend higher intakes of EFA. The Dietary Guidelines for Americans recommend 250 mg per day [21], and The European Safety Authority states that an additional 100–200 mg per day of DHA and beyond 250 mg per day of EPA + DHA [22].

Few studies have addressed the choline intake in women of childbearing age. Mygind et al. [23], in a sample of 125 New Zealand women of reproductive age (18–40), a mean intake of 316 mg/day, with only 16% of the sample meeting or exceeding the requirements. The mean intake of the overall childbearing-age women of ELANS was 305.0 mg/day with minimum and maximum values of 48.8 mg and 913.7 mg, respectively, with only 20% of women meeting or exceeding the established requirements.

Derbyshire et al. [20] and Vennemann et al. [22] found that choline intake in childbearing-age women in the USA, Europe, and Australia had a mean intake within a range of 244 to 443 mg/day, below the 425 mg recommended by the IOM considered for this study as the reference. Among the studies included in the literature revision by Derbyshire et al., only five percent of the non-pregnant women had choline intake above 631 mg/day in Sweden, above 578 mg/day in Finland, and above 543 mg/day in the Netherlands [20].

Regarding EFA consumption, Zhang et al. [24] found mean usual intakes of EPA and DHA combined from foods and dietary supplements of 88.1 + 3.0 mg per day, and 95% of their NHANES sample did not meet the intake of 250 mg per day. Nordgren et al. [6], in a sample of childbearing-age women and pregnant women, found no statistically different intakes of EPA and DHA, and the mean was 89.0 mg per day. Also, Nordgren et al. [6] reported that omega-3 fatty acid intake was significantly associated with poverty-to-income ratio, race, and educational attainment.

In our study, a difference in the intake of ALA and total omega-3 intake showed a slight increase as the educational level increased, but this was not observed for EPA and DHA, and women with basic or no education had the highest percentage of ALA deficiency and a lower proportion of excess in the diet. A contradictory difference was observed by marital status; while single women had the highest average intake of ALA and total omega-3, divorced or widowed had the lowest, and the opposite was found for EPA and DHA; divorced and widowed women showed higher intakes and lower intakes were reported for single women.

With the consistently reported “below of the requirements intake of choline and EPA and DHA” in women of fertile age, one could argue the following: first, women might not be aware of the existence and relevance of choline and EFA, and second, gynecologists and obstetricians are emphasizing educating women about other nutrients such as calcium, iron, folic acid, but not choline or EFA; third, a trend toward continuing existing eating habits could be expected, therefore, a childbearing-age woman with a low intake of choline, and/or inadequate, deficient or excessive, EFA is potentially at risk of inadequate intake of those nutrients when pregnant, especially when no intervention is implemented, and fourth, the lack of dietary diversity showed by women, which might not be enough for covering the choline and/or EFA requirements. Another concern at this point is the lack of choline supplements in the case of our study in the eight countries that constituted the ELANS study. Among the most-used prenatal supplements utilized in these countries, not even one contained choline in its composition, but omega-3 fatty acids were present.

In childbearing-age women, keeping a diverse and quality diet is key to achieving a good nutrition status [25]. Diet diversity and adequacy of micronutrients were studied previously in ELANS without considering choline and essential fatty acids [26]. Nevertheless, results show an important proportion of women (42.3%) consuming from 1–4 groups of foods (e.g., non-diverse diet), and women consuming five food groups (30.4%; acceptable diet). The most consumed foods, as per the 50% or more of the sample, were starchy staples (99.4%) and meat (84.2%), whereas eggs, an important source of choline, accounted only for 35.6% of women. Foods consumed the least were green leafy vegetables (6.8%) and nuts and seeds (2.8%). This provides a glimpse into food consumption in ELANS women of fertile age [26].

Usually, nutrition education emphasizes the relevance of iron and folic acid intake during pre-conception and pregnancy, and with good reason: they are important. In addition, adequate intake of other vitamins and minerals is recommended for pregnant women; therefore, calcium, vitamin A, and vitamin C, among others, are included in available OTC supplements. Also, emphasis has been made on adequate weight gain during pregnancy and the importance of eating proteins, the right fats, and carbohydrates, with subsequent recommendations to eat enough vegetables, fruits, lean proteins, and whole cereals. However, from the search among the available and most relevant health websites, one can infer that little information is given about choline intake and its sources. Important resources, including Johns Hopkins Medicine online [27], MedlinePlus [28], and even the WHO’s [29] online platforms do not mention choline and its importance during pregnancy in an explicit and friendly, accessible way or at other periods of life, nor the neuroprotective effects of EFAs such as AA or DHA in the developing brain and later in life [30]. The previous are resources with high traffic, and these tools might be a good way to promote and at least start catching the attention of women. The only webpage that included choline in a friendlier way was MyHealthfinder from the U.S. Department of Health and Human Services [31], but these online platforms are not popular with the Latin American population. Regretfully, webpage resources in Spanish are scarce, and because people in general do not know that choline and EFA are key factors to fetal brain development, they go unnoticed.

With the above being said, and how it is only recently that pre-conceptional health and nutrition status have been gaining attention as a part of the promotion of well-being to guarantee the health of future generations, the need for going beyond iron and folate has arrived. The pre-conceptional period has been neglected for some time. However, considering some of the concepts and theories that support the developmental origins of health and early exposure to diseases from environmental factors, we see that adequate nutrition is fundamental to good health before pregnancy. These nutritional determinants influence good health and play a role in making an eventual pregnancy successful, particularly when we know that 50% of pregnancies are non-planned [32].

Finally, an effort should be made to promote awareness about this important nutrient, as this study found no differences in women’s intake of choline by level of education. Therefore, it is possible that ELANS women of reproductive age might not have knowledge about the importance of choline intake, and their healthcare professional might not mention it or prescribe supplements to ensure adequate intake either by their diet or supplements, such as choline chloride, choline bitartrate, or phosphatidylcholine in consideration of choline’s role during the fetal period in the development of later cognitive abilities. Also, the fact that EFA has differences in educational and marital status for women should be considered [33]. Both findings provide windows of opportunity for intervention and education regarding these important nutrients for future generations in this region.

This study has several strengths, including the fact that every country has a national representative sample; there were two non-consecutive 24 h recalls, and the multiple pass method was used to minimize errors. However, it also has limitations, as this type of cross-sectional study does not allow for any causalities to be established, and only urban settings were considered, i.e., the rural population was not included. Also, we did not analyze the biochemical parameters of the participants nor analyze food chemistry characteristics; we used the available food composition tables described in the methods section. In addition, it should be clarified that the analyses of the available supplements in the region were based only on label information approved by the regulatory agencies of each country.

Finally, we wish to emphasize the importance of the pre-conceptional period for establishing new healthy routines and healthcare in young women, as it is a period for learning, promoting health, and introducing new positive changes into the lifestyle of Latin American women of childbearing age. We also want to suggest the inclusion of this period of life as an important period to be targeted with policies and programs that strengthen nutritional and health education as part of the interventions designed to ensure the well-being of future regional generations, as well as planning aid programs when needed that introduce foods that increase choline intake through diet, such as eggs and fish. This combination of food aid and education will become a sustainable component of education in parallel with the aid provided to those in need. Programs such as “One egg per day” for young children in Burkina Faso, where significant improvements were reported when this program was implemented, make sense if pre-conceptional and during-pregnancy deficiencies of these nutrients are found. Mothers reported more resilience and better psychomotor functioning in their young children after the consumption of one egg per day during the time of this project implementation [34]. Empowering women and promoting education and resilience through the policies and programs to be implemented at the regional level bring hope to better guarantee the well-being of future generations in Latin America.

## 5. Conclusions

Choline intake is inadequate for most ELANS women. Achieving the recommended intake through diet alone is challenging, especially when diets are not as diverse as they should be or meet the required quantities. Regarding EFA, the results show adequate and even excessive intakes for the studied population; however, the sources are different from those shown over other regions of the world. Regional efforts should be made to enhance adequate choline and EFA intake by improving the education of women and healthcare personnel through policies and programs aimed at guaranteeing the next generation’s health.

## Figures and Tables

**Figure 1 nutrients-16-03150-f001:**
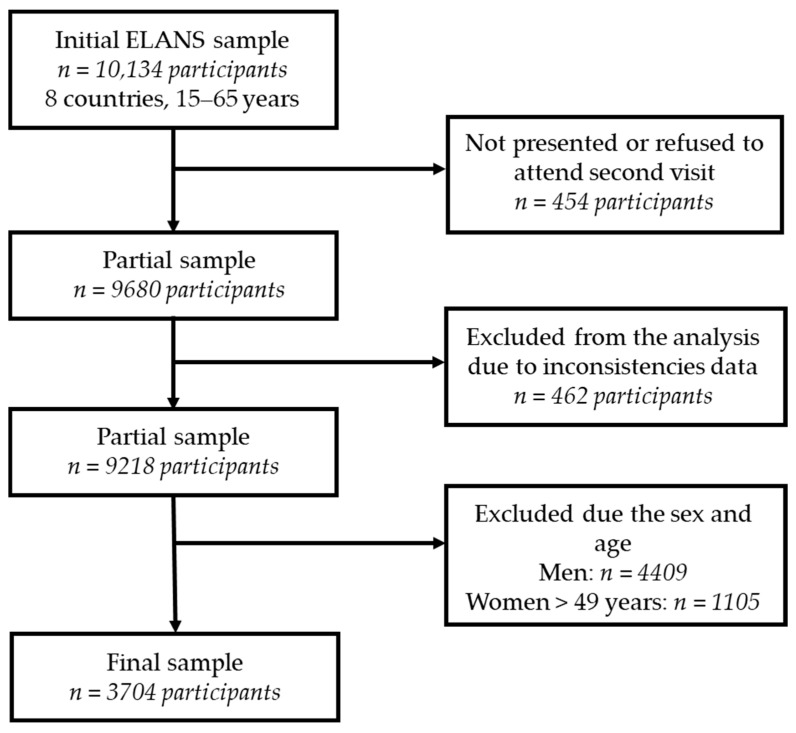
Flow chart of selection of ELANS childbearing-age women participants.

**Figure 2 nutrients-16-03150-f002:**
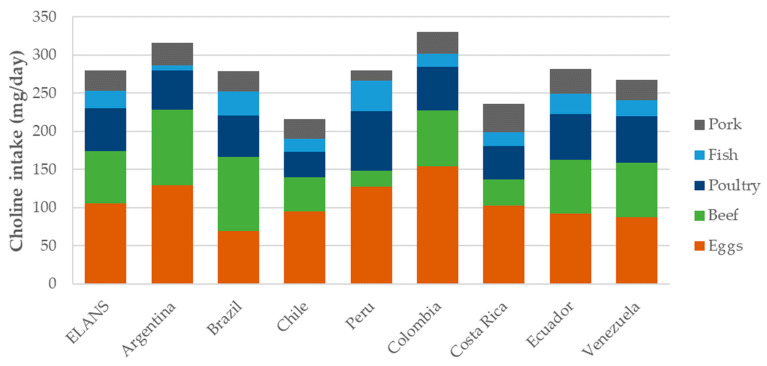
Mean daily choline intake (mg/day) by the five most contributing food groups and ELANS countries.

**Figure 3 nutrients-16-03150-f003:**
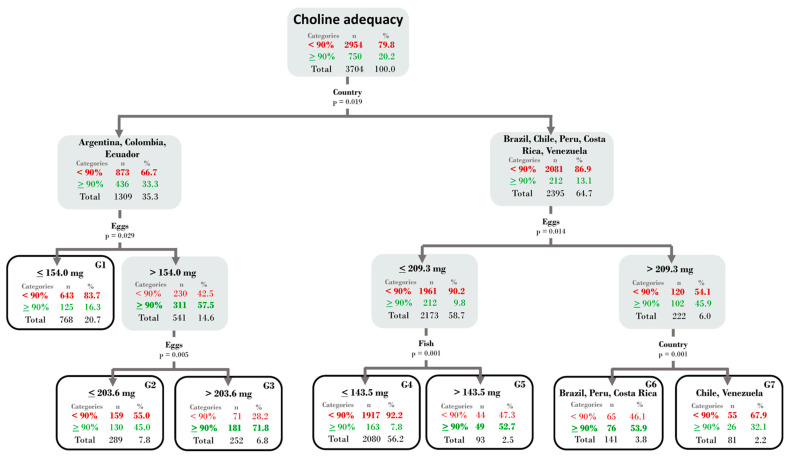
CRT Tree for choline adequacy in three levels. Each box presents the choline inadequacy (in red) and the adequate choline intake (in green). The interior nodes of the tree are represented with gray boxes and the terminal nodes with white boxes.

**Figure 4 nutrients-16-03150-f004:**
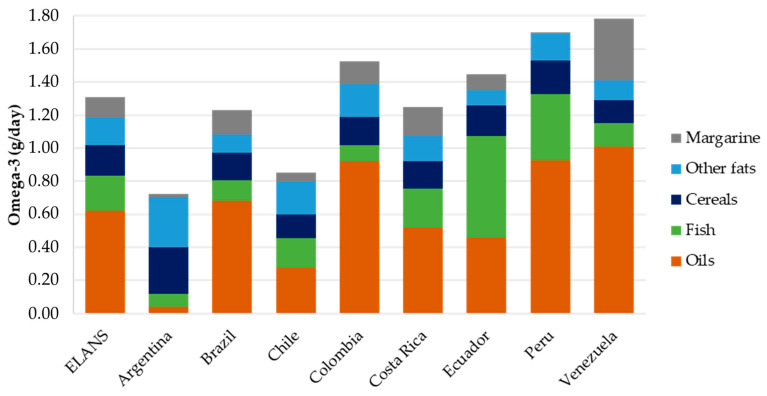
Mean daily omega-3 intake (g/day) by the five most contributing food groups and ELANS countries.

**Figure 5 nutrients-16-03150-f005:**
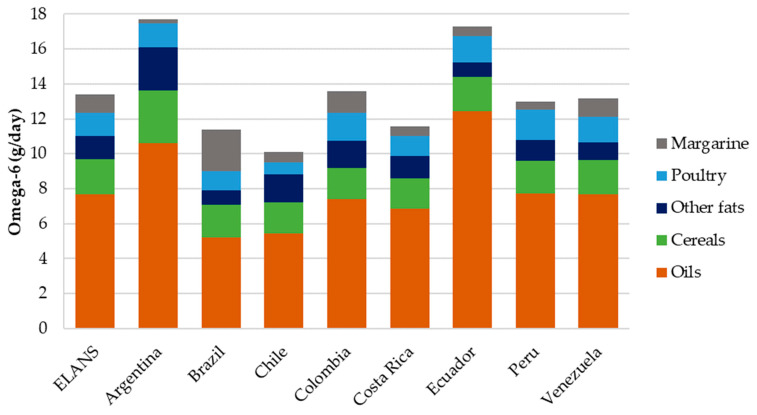
Mean daily omega-6 intake (g/day) by the five most contributing food groups and ELANS countries.

**Table 1 nutrients-16-03150-t001:** Sociodemographic characteristics of childbearing-age women aged 15–49 years in the Latin American Study of Nutrition and Health (ELANS).

Characteristics	ELANS		Argentina		Brazil		Chile		Colombia		Costa Rica		Ecuador		Peru		Venezuela	
	n	%	n	%	n	%	n	%	n	%	n	%	n	%	n	%	n	%
Sample size	3704	100.0	521	100.0	798	100.0	345	100.0	464	100.0	309	100.0	324	100.0	480	100.0	463	100.0
Age group																		
15–19 years	539	14.6	61	11.7	95	11.9	56	16.2	67	14.4	55	17.8	56	17.3	75	15.6	74	16.0
20–34 years	1771	47.8	244	46.8	379	47.5	152	44.1	224	48.3	143	46.3	152	46.9	246	51.3	231	49.9
35–49 years	1394	37.6	216	41.5	324	40.6	137	39.7	173	37.3	111	35.9	116	35.8	159	33.1	158	34.1
Socioeconomic status																		
Low	1982	53.5	271	52.0	381	47.8	150	43.5	314	67.7	115	37.2	164	50.7	235	49.0	352	76.0
Medium	1385	37.4	228	43.8	352	44.1	165	47.8	132	28.4	155	50.2	120	37.0	154	32.0	79	17.1
High	337	9.1	22	4.2	65	8.1	30	8.7	18	3.9	39	12.6	40	12.3	91	19.0	32	6.9
Education level																		
None and basic	2139	57.7	359	68.9	344	43.1	217	62.9	280	60.4	248	80.3	271	83.6	111	23.1	309	66.8
High school	1196	32.3	134	25.7	386	48.4	91	26.4	132	28.4	40	12.9	35	10.8	320	66.7	58	12.5
Bachelor’s degree	369	10.0	28	5.4	68	8.5	37	10.7	52	11.2	21	6.8	18	5.6	49	10.2	96	20.7
Marital Status																		
Single	1527	41.2	186	35.7	342	42.9	163	47.2	206	44.4	128	41.4	111	34.3	168	35.0	223	48.2
Married/Couple	1863	50.3	271	52.0	388	48.6	162	47.0	226	48.7	154	49.8	187	57.7	276	57.5	199	43.0
Divorced/Widowed	314	8.5	64	12.3	68	8.5	20	5.8	32	6.9	27	8.7	26	8.0	36	7.5	41	8.9

**Table 2 nutrients-16-03150-t002:** Choline consumption parameters of childbearing-age women aged 15–49 years in the Latin American Study of Nutrition and Health (ELANS).

		Choline Consumption				Choline Adequacy				Percentage of Sample *					
Characteristics		mg/day				Percentage (%)				Deficient		Normal		Excessive	
	n	Mean	SEM	Min	Max	Mean	SEM	Min	Max	n	%	n	%	n	%
Country															
ELANS	3704	305.0	1.6	48.8	913.7	71.8	0.4	11.5	215.0	2954	79.8	520	14.0	230	6.2
Argentina	521	338.7	4.4	100.3	912.6	79.7	1.0	23.6	214.7	375	72.0	96	18.4	50	9.6
Brazil	798	294.7	3.5	48.8	913.7	69.3	0.8	11.5	215.0	666	83.5	92	11.5	40	5.0
Chile	345	238.6	3.8	62.1	503.7	56.1	0.9	14.6	118.5	329	95.4	14	4.1	2	0.6
Colombia	464	373.1	4.8	138.4	697.1	87.8	1.1	32.6	164.0	256	55.2	120	25.9	88	19.0
Costa Rica	309	262.5	4.9	69.0	662.9	61.8	1.1	16.2	156.0	280	90.6	21	6.8	8	2.6
Ecuador	324	327.6	5.0	138.4	614.8	77.1	1.2	32.6	144.7	242	74.7	63	19.4	19	5.9
Peru	480	313.8	3.7	99.8	663.9	73.8	0.9	23.0	156.2	382	79.6	80	16.7	18	3.8
Venezuela	463	269.8	3.6	69.2	542.0	63.5	0.8	16.3	127.5	424	91.6	34	7.3	5	1.1
Age group															
15–19 years	539	301.3	4.1	69.0	662.9	70.9	1.0	16.2	156.0	441	81.8	66	12.2	32	5.9
20–34 years	1771	310.3	2.3	87.0	912.6	73.0	0.6	20.5	214.7	1381	78.0	277	15.6	113	6.4
35–49 years	1394	299.9	2.7	48.8	913.7	70.6	0.6	11.5	215.0	1132	81.2	177	12.7	85	6.1
Socioeconomic status															
Low	1982	306.5	2.3	48.8	913.7	72.1	0.5	11.5	215.0	1564	78.9	276	13.9	142	7.2
Medium	1385	303.1	2.6	69.2	912.6	71.3	0.6	16.3	214.7	1118	80.7	190	13.7	77	5.6
High	337	304.3	4.7	136.1	584.3	71.6	1.1	32.0	137.5	272	80.7	54	16.0	11	3.3
Education level															
None and basic	2139	301.4	2.2	48.8	913.7	70.9	0.5	11.5	215.0	1708	79.9	290	13.6	141	6.6
High school	1196	311.2	2.8	87.0	697.1	73.2	0.6	20.5	164.0	954	79.8	167	14.0	75	6.3
Bachelor’s degree	369	305.9	4.6	136.1	584.3	72.0	1.1	32.0	137.5	292	79.1	63	17.1	14	3.8
Marital Status															
Single	1527	301.5	2.5	69.0	913.7	70.9	0.6	16.2	215.0	1238	81.1	199	13.0	90	5.9
Married/Couple	1863	308.9	2.3	48.8	912.6	72.7	0.5	11.5	214.7	1459	78.3	283	15.2	121	6.5
Divorced/Widowed	314	299.5	5.5	62.1	637.1	70.5	1.3	14.6	149.9	257	81.8	38	12.1	19	6.1

* This data refer to the number and row percentages (%) of women with deficient, normal, or excessive choline intake. SEM—Standard error of the mean. Min—Minimum value. Max—Maximum value.

**Table 3 nutrients-16-03150-t003:** Alpha-linolenic acid consumption parameters of childbearing-age women aged 15–49 years in the Latin American Study of Nutrition and Health (ELANS).

		Alpha-Linolenic Acid				Alpha-Linolenic Acid Adequacy				Percentage of Sample *					
Characteristics		g/day				Percentage (%)				Deficient		Normal		Excessive	
	n	Mean	SEM	Min	Max	Mean	SEM	Min	Max	n	%	n	%	n	%
Country															
ELANS	3704	1.38	0.01	0.23	7.71	125.5	0.9	20.7	701.2	1049	28.3	597	16.1	2058	55.6
Argentina	521	0.98	0.02	0.29	2.89	89.1	1.6	26.3	262.6	300	57.6	88	16.9	133	25.5
Brazil	798	1.44	0.02	0.38	4.79	130.9	1.8	34.5	435.3	159	19.9	131	16.4	508	63.7
Chile	345	1.01	0.02	0.23	3.10	92.2	2.2	20.7	282.0	192	55.7	59	17.1	94	27.2
Colombia	464	1.63	0.03	0.45	7.71	148.5	2.9	40.6	701.2	61	13.1	55	11.9	348	75.0
Costa Rica	309	1.30	0.03	0.41	3.93	118.0	2.8	37.3	357.3	98	31.7	56	18.1	155	50.2
Ecuador	324	1.23	0.03	0.39	4.05	111.4	2.7	35.0	368.6	123	38.0	76	23.5	125	38.6
Peru	480	1.56	0.03	0.50	4.35	142.2	2.4	45.1	395.5	61	12.7	76	15.8	343	71.5
Venezuela	463	1.72	0.03	0.30	5.01	156.3	2.9	26.9	455.6	55	11.9	56	12.1	352	76.0
Age group															
15–19 years	539	1.49	0.03	0.34	5.01	135.6	2.7	31.0	455.6	132	24.5	78	14.5	329	61.0
20–34 years	1771	1.42	0.02	0.29	7.71	129.1	1.4	26.3	701.2	465	26.3	284	16.0	1022	57.7
35–49 years	1394	1.29	0.01	0.23	4.35	116.9	1.3	20.7	395.5	452	32.4	235	16.9	707	50.7
Socioeconomic status															
Low	1982	1.39	0.01	0.23	7.71	126.8	1.3	20.7	701.2	554	28.0	318	16.0	1110	56.0
Medium	1385	1.35	0.02	0.29	4.68	122.9	1.4	26.3	425.4	407	29.4	218	15.7	760	54.9
High	337	1.41	0.03	0.39	3.64	128.6	3.0	35.0	330.7	88	26.1	61	18.1	188	55.8
Education level															
None and basic	2139	1.32	0.01	0.23	7.71	120.2	1.2	20.7	701.2	698	32.6	365	17.1	1076	50.3
High school	1196	1.45	0.02	0.33	4.79	132.0	1.6	29.6	435.3	273	22.8	182	15.2	741	62.0
Bachelor’s degree	369	1.48	0.03	0.30	3.58	134.9	2.8	27.3	325.1	78	21.1	50	13.6	241	65.3
Marital Status															
Single	1527	1.42	0.02	0.34	4.79	129.3	1.5	31.0	435.3	406	26.6	242	15.8	879	57.6
Married/Couple	1863	1.36	0.01	0.23	7.71	123.5	1.3	20.9	701.2	549	29.5	295	15.8	1019	54.7
Divorced/Widowed	314	1.31	0.03	0.23	3.45	119.2	2.9	20.7	313.8	94	29.9	60	19.1	160	51.0

* This data refer to the number and row percentages (%) of women with deficient, normal, or excessive choline intake. SEM—Standard error of the mean. Min—Minimum value. Max—Maximum value.

**Table 4 nutrients-16-03150-t004:** Other omega-3 fatty acids intake of childbearing women aged 15–49 years in the Latin American Study of Nutrition and Health (ELANS).

		Eicosapentaenoic Acid				Docosahexaenoic Acid				Omega-3 Total			
Characteristics		mg/day				mg/day				g/day			
	n	Mean	SEM	Min	Max	Mean	SEM	Min	Max	Mean	SEM	Min	Max
Country													
ELANS	3704	21.9	0.3	1.2	275.3	71.2	0.9	6.1	814.5	1.47	0.01	0.25	7.80
Argentina	521	16.6	0.4	2.0	92.5	63.0	1.6	7.6	326.5	1.06	0.02	0.33	2.96
Brazil	798	16.9	0.4	1.2	138.8	59.3	1.5	6.1	491.7	1.52	0.02	0.45	4.97
Chile	345	18.5	1.1	1.9	171.7	64.3	3.2	6.6	479.5	1.10	0.02	0.25	3.15
Colombia	464	19.7	0.7	2.5	203.6	62.9	1.7	6.5	357.3	1.72	0.03	0.47	7.80
Costa Rica	309	21.4	0.9	3.4	97.7	70.8	2.6	14.7	332.1	1.39	0.03	0.46	3.99
Ecuador	324	37.0	2.0	7.3	275.3	113.5	5.8	7.1	814.5	1.38	0.03	0.46	4.39
Peru	480	31.1	1.1	4.7	151.1	89.6	2.7	16.3	346.5	1.68	0.03	0.54	4.44
Venezuela	463	21.1	0.5	6.6	99.9	66.4	1.6	17.0	248.4	1.81	0.03	0.40	5.12
Age group													
15–19 years	539	20.7	0.7	2.4	197.9	68.0	2.3	8.5	549.8	1.58	0.03	0.38	5.12
20–34 years	1771	22.0	0.5	1.2	275.3	71.0	1.3	6.1	814.5	1.51	0.02	0.33	7.80
35–49 years	1394	22.1	0.5	1.9	216.1	72.9	1.6	6.6	621.8	1.38	0.02	0.25	4.44
Socioeconomic status													
Low	1982	21.9	0.4	2.0	275.3	71.3	1.3	6.1	814.5	1.49	0.01	0.25	7.80
Medium	1385	21.2	0.5	1.2	203.6	70.2	1.4	7.1	479.5	1.44	0.02	0.33	4.85
High	337	24.1	1.2	2.4	195.6	75.2	3.3	13.8	548.8	1.51	0.03	0.45	3.69
Education level													
None and basic	2139	21.6	0.4	1.2	216.1	71.5	1.2	6.6	621.8	1.42	0.01	0.25	7.80
High school	1196	22.0	0.6	2.3	275.3	70.6	1.6	6.1	814.5	1.55	0.02	0.42	4.97
Bachelor’s degree	369	22.8	1.1	2.0	203.6	72.0	2.8	6.5	357.3	1.58	0.03	0.37	3.67
Marital Status													
Single	1527	20.6	0.4	1.9	197.9	67.1	1.3	6.1	549.8	1.51	0.02	0.38	4.97
Married/Couple	1863	22.7	0.5	1.2	275.3	73.9	1.4	8.5	814.5	1.45	0.01	0.33	7.80
Divorced/Widowed	314	23.0	1.3	2.7	216.1	75.4	3.7	7.1	621.8	1.41	0.03	0.25	3.54

**Table 5 nutrients-16-03150-t005:** Linoleic acid consumption parameters of childbearing-age women aged 15–49 years in the Latin American Study of Nutrition and Health (ELANS).

		Linoleic Acid				Linoleic Acid Adequacy				Percentage of Sample *					
Characteristics		g/day				Percentage (%)				Deficient		Normal		Excessive	
	n	Mean	SEM	Min	Max	Mean	SEM	Min	Max	n	%	n	%	n	%
Country															
ELANS	3704	13.91	0.09	2.94	57.92	125.5	0.9	20.7	701.3	1131	30.5	732	19.8	1841	49.7
Argentina	521	15.11	0.26	3.63	41.28	89.1	1.6	26.3	262.7	132	25.3	94	18.0	295	56.6
Brazil	798	12.84	0.18	3.22	36.28	130.9	1.8	34.5	435.3	301	37.7	166	20.8	331	41.5
Chile	345	11.44	0.21	2.94	27.21	92.2	2.2	20.7	282.0	163	47.2	70	20.3	112	32.5
Colombia	464	13.94	0.26	3.58	57.92	148.5	2.9	40.6	701.3	134	28.9	98	21.1	232	50.0
Costa Rica	309	13.44	0.27	3.65	31.18	118.0	2.8	37.3	357.3	99	32.0	71	23.0	139	45.0
Ecuador	324	17.47	0.35	5.88	43.75	111.5	2.7	35.0	368.6	38	11.7	48	14.8	238	73.5
Peru	480	14.10	0.22	4.75	34.53	142.2	2.4	45.1	395.5	118	24.6	98	20.4	264	55.0
Venezuela	463	13.81	0.25	3.93	39.87	156.3	2.9	26.9	455.6	146	31.5	87	18.8	230	49.7
Age group															
15–19 years	539	15.19	0.26	4.36	41.88	135.6	2.7	31.0	455.6	99	18.4	99	18.4	341	63.3
20–34 years	1771	14.18	0.13	3.58	57.92	129.1	1.4	26.3	701.2	522	29.5	347	19.6	902	50.9
35–49 years	1394	13.06	0.13	2.94	36.28	116.9	1.3	20.7	395.5	510	36.6	286	20.5	598	42.9
Socioeconomic status															
Low	1982	13.95	0.12	3.04	57.92	126.8	1.3	20.7	701.2	598	30.2	391	19.7	993	50.1
Medium	1385	13.89	0.14	2.94	41.28	122.9	1.4	26.3	425.4	423	30.5	271	19.6	691	49.9
High	337	13.74	0.30	4.45	43.75	128.6	3.0	35.0	330.7	110	32.6	70	20.8	157	46.6
Education level															
None and basic	2139	13.99	0.12	2.94	57.92	120.2	1.2	20.7	701.2	665	31.1	409	19.1	1065	49.8
High school	1196	13.90	0.15	4.03	47.77	132.0	1.6	29.6	435.3	351	29.3	249	20.8	596	49.8
Bachelor’s degree	369	13.44	0.24	3.58	31.37	134.9	2.8	27.3	325.1	115	31.2	74	20.1	180	48.8
Marital Status															
Single	1527	14.07	0.14	3.58	47.77	129.3	1.5	31.0	435.3	449	29.4	293	19.2	785	51.4
Married/Couple	1863	13.84	0.13	2.94	57.92	123.5	1.3	20.9	701.2	578	31.0	367	19.7	918	49.3
Divorced/Widowed	314	13.52	0.29	3.04	35.84	119.2	2.9	20.7	313.8	104	33.1	72	22.9	138	43.9

* This data refer to the number and row percentages (%) of women with deficient, normal, or excessive choline intake. SEM—Standard error of the mean. Min—Minimum value. Max—Maximum value.

**Table 6 nutrients-16-03150-t006:** Other omega-6 fatty acids intake of childbearing-age women aged 15–49 years in the Latin American Study of Nutrition and Health (ELANS).

		Arachidonic Acid				Omega-6 Total			
Characteristics		mg/day				g/day			
	n	Mean	SEM	Min	Max	Mean	SEM	Min	Max
Country									
ELANS	3704	141.9	0.9	22.0	536.8	14.0	0.1	3.0	58.1
Argentina	521	146.9	2.5	23.2	372.3	15.3	0.3	3.7	41.5
Brazil	798	126.2	1.8	26.3	350.3	13.0	0.2	3.3	36.4
Chile	345	116.3	2.7	29.4	482.8	11.6	0.2	3.0	27.3
Colombia	464	149.8	2.5	22.0	335.7	14.1	0.3	3.6	58.1
Costa Rica	309	133.2	3.0	25.3	415.2	13.6	0.3	3.7	31.3
Ecuador	324	172.9	3.6	53.4	536.8	17.6	0.4	6.0	44.1
Peru	480	157.7	2.4	23.2	321.0	14.3	0.2	4.8	34.8
Venezuela	463	142.4	2.2	50.9	300.1	13.9	0.3	4.0	40.1
Age group									
15–19 years	539	141.6	2.4	25.3	343.8	15.3	0.3	4.4	42.0
20–34 years	1771	144.7	1.3	26.3	536.8	14.3	0.1	3.6	58.1
35–49 years	1394	138.5	1.5	22.0	482.8	13.2	0.1	3.0	36.4
Socioeconomic status									
Low	1982	143.9	1.2	22.0	536.8	14.1	0.1	3.1	58.1
Medium	1385	138.8	1.5	23.2	415.2	14.0	0.1	3.0	41.5
High	337	143.3	3.1	23.2	348.6	13.9	0.3	4.5	44.1
Education level									
None and basic	2139	141.7	1.2	22.0	482.8	14.1	0.1	3.0	58.1
High school	1196	143.6	1.6	24.3	536.8	14.0	0.2	4.1	48.0
Bachelor’s degree	369	137.7	2.6	23.2	340.5	13.6	0.2	3.6	31.5
Marital Status									
Single	1527	139.3	1.4	23.2	415.2	14.2	0.1	3.6	48.0
Married/Couple	1863	144.9	1.3	26.3	536.8	14.0	0.1	3.0	58.1
Divorced/Widowed	314	137.0	3.3	22.0	368.3	13.7	0.3	3.1	36.0

SEM—Standard error of the mean. Min—Minimum value. Max—Maximum value.

**Table 7 nutrients-16-03150-t007:** Choline and essential fatty acids are the three most prominent prenatal supplements by country.

		Pre-Natal Supplements		
Country		Option 1	Option 2	Option 3
Argentina	Commercial name	Maternil	Supradyn	---
	Choline (mg)	0	0	---
	Omega-3 (mg)	135 (105 DHA)	0	---
	Omega-6 (mg)	0	0	0
Brazil	Commercial name	Materna	Regenesis	OragestanGold
	Choline (mg)	0	0	0
	Omega-3 (mg)	0	300 (250 DHA)	300 (200 DHA)
	Omega-6 (mg)	0	0	0
Chile	Commercial name	Enfamom	Supradyn	One-a-day
	Choline (mg)	0	0	0
	Omega-3 (mg)	235 (200 DHA)	0	235 (200 DHA)
	Omega-6 (mg)	0	0	0
Colombia	Commercial name	Enfamom	Natele	One-a-day
	Choline (mg)	0	0	0
	Omega-3 (mg)	235 (200 DHA)	125 (125 DHA)	235 (200 DHA)
	Omega-6 (mg)	0	0	0
Costa Rica	Commercial name	Gestavit	Natele	Medox
	Choline (mg)	0	0	0
	Omega-3 (mg)	150 (150 DHA)		
	Omega-6 (mg)	0	0	0
Ecuador	Commercial name	Natalben	Natele	Panvit
	Choline (mg)	0	0	0
	Omega-3 (mg)	240 (200 DHA)	0	0
	Omega-6 (mg)	0	0	0
Perú	Commercial name	Gestavit	Supradyn	Maddre
	Choline (mg)	0	0	0
	Omega-3 (mg)	150 (150 DHA)	0	0
	Omega-6 (mg)	0	0	0
Venezuela	Commercial name	Maternavit	Miagest	Multivinol
	Choline (mg)	0	0	0
	Omega-3 (mg)	275	320	0
	Omega-6 (mg)	0	0	0

## Data Availability

The data presented in this study are available upon request from the corresponding author. The data are not publicly available due to privacy or ethical restrictions.

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
