# Peer review of "Inadequate Intake of Choline and Essential Fatty Acids in Latin American Childbearing-Age Women as a Regional Pre-Conceptional Disadvantage: ELANS Results"

_nutrients, 2024, doi:10.3390/nu16183150_

Round 1
Reviewer 1 Report
Comments and Suggestions for Authors
The work of Herrera-Cuenca Choline and essential fatty acids inadequate intake in Latin American childbearing age women as a regional pre-conceptional disadvantage. ELANS results
presents mathematical application to monitorng the appropriate quantity intake of choline and fatty acid in woman in Latin America, without ANALITICAL CHARACTERIZATION of FOODs ; in order to present these data they should make the analysis of the food (for example HPLC , GC) of each region and then they can apply the alghoritm for study the correlations and all statistical analysises.
Therefore this is just a mathematical study that is not suitable for this journal, I suggest a specic mathematical jourmal like algorithms of MDPI
No comments
Author Response
Thank you for this interesting comment that our team fully discussed. We respectfully disagree, with submitting to a math journal.
Our analysis is grounded in the application of standard data analysis procedures, commonly found in modern statistics textbooks. Maybe the inclusion of the word “algorithm” causes this misleading, so we have deleted the word and describe the statistical procedure as a classification tree (CT).
It is important to clarify that publications in the field of mathematics typically aim to introduce or exemplify innovative methodologies. In contrast, the methodology used in this article adheres to standard practices, making it less relevant to the mathematics community. However, the results are highly relevant to the fields of nutrition, health, and epidemiology, as evidenced by the over 80 downloads of the pre-print file. We believe this article makes a significant contribution to the nutritional sciences, particularly since, to the best of our knowledge, this is the first epidemiological study to analyze choline and essential fatty acid intake among women of reproductive age across eight countries in Latin America.
Reviewer 2 Report
Comments and Suggestions for Authors
The manuscript by Herrera-Cuenca et al. reports on choline and essential fatty acids intake in south american childbearing age women. This is an interesting study relating with newborns health, especially regarding neural abilities. In that respect, the different countries in South America are compared. However, the data taken into account are only from food, not from nutrient molecules. This is a weakness, as we know that the biochemical forms of EFA for brain intake for instance is important, with a large preference of ARA and DHA brain uptake from blood lysophosphatidylcholine compared to free FA. So ARA and or DHA from dietary phosphatidylcholine are more available to brain than from triglycerides. This should be taken into consideration.
Specific comments:
- When reporting about choline intake, it would be helpfull to define its origin, whether it can be provided from phospholipids (particularly phosphatidylcholine/PC), through phospholipase D cleavage for instance. This would be especially relevant when related with egg or fish intake, as PC from eggs is rich in LA while PC from fish should contain substantial amounts of EPA and DHA.
- When considering omega-3 PUFA, it is important to remember that EPA and DHA produced from ALA in the body is quite low. Bringing DHA within fish is then much more efficient for brain accumulation than from ALA in dietary triglycerides (in oils/margarines).
- The general discussion including other nutrients in food, such as vitamins and iron, just dilute messages relating to choline and EFA.
Author Response
Specific comments:
- When reporting about choline intake, it would be helpfull to define its origin, whether it can be provided from phospholipids (particularly phosphatidylcholine/PC), through phospholipase D cleavage for instance. This would be especially relevant when related with egg or fish intake, as PC from eggs is rich in LA while PC from fish should contain substantial amounts of EPA and DHA.
- When considering omega-3 PUFA, it is important to remember that EPA and DHA produced from ALA in the body is quite low. Bringing DHA within fish is then much more efficient for brain accumulation than from ALA in dietary triglycerides (in oils/margarines).
Answer:
Thank you for your comment. We would like to emphasize that this study is epidemiological in nature, and the initial objectives did not include chemical or biomolecular analyses, among other reasons because the budget for that in the large sample of this study would have made it impossible to cover. The data on food intake was obtained using 24-hour recalls and the multiple pass method, both of which are described in the methods section. The compositional analysis was based on food composition available through the software NDSR and the food composition tables from each country to address recipes not included in the system. A standardization methodology was applied across the entire analysis. It is indeed a limitation of our study that we did not include biochemical or molecular analyses, either in the participants (blood sample) or the foods (further chemical composition analyses). This limitation has been introduced in the study's limitations section. Lines 576-579.
The food sources of choline were identified based on reported food intake, and due to the high prevalence of inadequacy, a categorization tree (CT) was applied to determine when choline intake was higher. Our aim is to support the promotion of choline-rich foods, such as eggs and fish, which we highlighted in the results, and to enhance awareness among this population group, as pre-conceptional nutrition education.
- The general discussion including other nutrients in food, such as vitamins and iron, just dilute messages relating to choline and EFA.
Thank you for this valuable comment. This brings attention to what has been done to some micronutrients that can be a role model to follow in the educational path toward a better choline and EFA intake. We briefly mention at the introduction and the discussion as a way to exemplify what has been done, and what we can do for these nutrients, relatively unknown for the general population, particularly in Latin America.
Reviewer 3 Report
Comments and Suggestions for Authors
This eminent article examines the intake of choline (vitamin B4) and EFA (vitamin F) in young women, before getting pregnant. This multi-center study is comprehensive and complete. I have some remarks:
Results: the tables are clear, but none of them seems to mention the relative omega-6 : omega-3 ratio, as only absolute quantities are shown. Nevertheless, this ratio is important concerning health issues.
Statistics: the Chi-square test could be complemented by Yates' correction.
Comments on the Quality of English LanguageSome typo's throughout the text.
Author Response
Results: the tables are clear, but none of them seems to mention the relative omega-6 : omega-3 ratio, as only absolute quantities are shown. Nevertheless, this ratio is important concerning health issues.
Statistics: the Chi-square test could be complemented by Yates' correction.
Answer:
Thank you for this valuable comment. We have included a table as supplementary material and a description of the results. Lines 416-424
We want to clarify, that Yates’ correction would not apply to this sample as it is not as small as it should be to apply it. Yates’ correction is used when small samples are to be analyzed.
Reviewer 4 Report
Comments and Suggestions for Authors
- line 34: what age exactly?
- line 36,38, 42, 112, 185, 195, 205: EFA
- line 50: any source to back up this statement?
- Introduction: How about combining the paragraphs about choline?
- lines 57-58: The cited publications state the association of choline with methylation, there is no mention of repair or synthesis. Please provide sources to support these statements.
-line 64: physical performance in adults?in children? Please specify
- line 66: no prior information on lactation
- line 102, 111: Please add an explanation of the abbreviation
- line 118: "thus our" - Isn't the continuation part of the sentence missing here?
- Figures 1, 3, and 4: The figure should appear after it is mentioned in the text, not before
- line 133: excess space?
- line 175: two groups of EFA rather than two fatty acids
- line 192: Why exactly was 90% taken as the cutoff point?
- section 2,4: Has the content of the tested ingredients been confirmed?
- line 231: Is there a list of participating institutions available somewhere?
- line 234: Which system?
- line 239: abbreviation has been introduced before
- Table 1: Is it impossible to present names of characteristics in one line?
- it looks like there is a problem with page numbering on the entire document
- Table 2: Is it impossible to present the names of characteristics in one line? no explanation for "*"
- Figure 2: shouldn't the description of the Y axis be mg/day
- Table 3, 4: putting the table on one page will make it easier to read, statistically significant differences discussed in the text should be highlighted in the table
- Figure 4, 5: [g/day] on Y axis
- Table 7: table should appear before starting a new section, title on the same page as the table will make it very easy to read
- line 469: a paragraph about Choline and the sentence is about EFA. Is this the right paragraph?
- line 475, 478, 483: "omega 3" - in earlier sections it was written as omega-3
- line 477: "6" - is this a reference?
Author Response
Thank you for all your valuable comments that enrich our manuscript
- line 34: what age exactly? We included the ages (15-49 years)
- line 36,38, 42, 112, 185, 195, 205: EFA Done, we have changed it.
- line 50: any source to back up this statement? We eliminated this statement
- Introduction: How about combining the paragraphs about choline? We edited the paragraphs. Lines 50-63.
- lines 57-58: The cited publications state the association of choline with methylation, there is no mention of repair or synthesis. Please provide sources to support these statements. We included the functions of synthesis and repair, which are supported by the references. “…DNA methylation processes, synthesis and repair” Lines 54-55.
-line 64: physical performance in adults? in children? Please specify.
Answer: We clarify that we referred to adults (9 of 14 studies were done in animals, 3 in vitro, 1 in elite athletes and 1 in menopause women). Line 57
- line 66: no prior information on lactation. Lactation was included in line 60.
- line 102, 111: Please add an explanation of the abbreviation. Resolved. “Food and Agriculture Organization of the United Nations (FAO)” Lines 99-100. And, “Latin American Study of Nutrition and Health (ELANS) by its acronym in Spanish (Estudio Latinoamericano de Nutrición y Salud)” Lines 109-110.
- line 118: "thus our" - Isn't the continuation part of the sentence missing here? … … aim was to study the choline and essential fatty acids intake in childbearing age women, identify the food sources, if supplements that contained choline and EFA existed across the studied countries and identify what potential adequate policies would help if deficiencies were found.
Agree. It was a formatting problem, we resolved it. “thus our aim was to study the choline and EFA intake in childbearing age women, identify the food sources, if supplements that contained choline and EFA existed across the studied countries and identify what potential adequate policies would help if defi-ciencies were found.” Lines 119-122.
- Figures 1, 3, and 4: The figure should appear after it is mentioned in the text, not before. Answer:
We understand the importance of properly formatting and positioning figures in a research publication. However, I must mention that final formatting, including decisions about figure placements, is often done by the journal's design team. They are experienced in creating a clear and easy to navigate layout that's suitable for both print and digital versions of the article.
- line 133: excess space?
Answer: Highlighted for adequate formatting. The journal's design team could help on this.
- line 175: two groups of EFA rather than two fatty acids. Resolved. Line 187878
- Line 192: Why exactly was 90% taken as the cutoff point?.
Answer:
In accordance with the researcher’s agreement, we chose to define adequate intakes using a range of 90-110% for this study and other ELANS studies. This range was selected because the probability of adequacy is higher compared to using lower cut-off points. Other researchers and experts have also adopted this approach, finding that adequate intake is more likely when values between 75-100% of the intake are used.
-section 2,4: Has the content of the tested ingredients been confirmed?
Answer:
We based our supplements analyses on labels already approved by regulatory agencies on each country and it is a limitation of this study as we cannot go further from this level. We did not have the objective of performing chemical or biomolecular analysis on these supplements, this is just to get an idea of what is available or not in the region. This limitation was added on lines 582-584.
- line 231: Is there a list of participating institutions available somewhere?
Answer:
We are concerned that listing all the participating institutions might be cumbersome for the reader. The institutions involve are listed in the affiliations of the authors, and in the acknowledgement section.
- line 234: Which system? We eliminated this mention.
- line 239: abbreviation has been introduced before. Resolved
- Table 1: Is it impossible to present names of characteristics in one line? Resolved.
- it looks like there is a problem with page numbering on the entire document. Will be taken care by the journal’ designing team.
- Table 2: Is it impossible to present the names of characteristics in one line? no explanation for "*" Resolved on our end, however the journal’s designing team should be the one in charged for this
- Figure 2: shouldn't the description of the Y axis be mg/day. Resolved
- Table 3, 4: putting the table on one page will make it easier to read, statistically significant differences discussed in the text should be highlighted in the table. The designer needs to try to get in one page the table, but on the other side adding columns to highlight the significant differences may require more space.
- Figure 4, 5: [g/day] on Y axis Resolved
- Table 7: table should appear before starting a new section, title on the same page as the table will make it very easy to read. The designer needs to take care of this.
- line 469: a paragraph about Choline and the sentence is about EFA. Is this the right paragraph? Answer: We re-phrased this for better understanding. Lines 478-487.
- line 475, 478, 483: "omega 3" - in earlier sections it was written as omega-3. Resolved
- line 477: "6" - is this a reference? Resolved